# Adaptive and non-adaptive divergence in a common landscape

Joost A.M. Raeymaekers [1,2,3], Anurag Chaturvedi[1,4], Pascal I. Hablützel [1,5], Io Verdonck[1], Bart Hellemans[1], Gregory E. Maes [1,6,7], Luc De Meester[4] & Filip A.M. Volckaert[1]

Species in a common landscape often face similar selective environments. The capacity of organisms to adapt to these environments may be largely species specific. Quantifying shared and unique adaptive responses across species within landscapes may thus improve our understanding of landscape-moderated biodiversity patterns. Here we test to what extent populations of two coexisting and phylogenetically related fishes—three-spined and nine-spined stickleback—differ in the strength and nature of neutral and adaptive divergence along a salinity gradient. Phenotypic differentiation, neutral genetic differentiation and genomic signatures of adaptation are stronger in the three-spined stickleback. Yet, both species show substantial phenotypic parallelism. In contrast, genomic signatures of adaptation involve different genomic regions, and are thus non-parallel. The relative contribution of spatial and environmental drivers of population divergence in each species reflects different strategies for persistence in the same landscape. These results provide insight in the mechanisms underlying variation in evolutionary versatility and ecological success among species within landscapes.

[1] Laboratory of Biodiversity and Evolutionary Genomics, KU Leuven, B-3000 Leuven, Belgium. [2] Centre for Biodiversity Dynamics, Department of Biology, Norwegian University of Science and Technology, N-7491 Trondheim, Norway. [3] Faculty of Biosciences and Aquaculture, Nord University, N-8049 Bodø, Norway. [4] Laboratory of Aquatic Ecology, Evolution and Conservation, KU Leuven, B-3000 Leuven, Belgium. [5] Flanders Marine Institute, B-8400 Oostende, Belgium. [6] Genomics Core, Center for Human Genetics, UZ Leuven, B-3000 Leuven, Belgium. [7] Centre for Sustainable Tropical Fisheries and Aquaculture, Comparative Genomics Centre, College of Science and Engineering, James Cook University, Townsville, QLD 4811, Australia. Joost A.M. Raeymaekers and Anurag Chaturvedi contributed equally to this work. Correspondence and requests for materials should be addressed to J.A.M.R. (email: joostraeymaekers@gmail.com)

Why natural populations do or do not diversify in heterogeneous landscapes is central to the understanding of processes generating and maintaining biological diversity. The answer lies in a combination of factors, including the strength of selection relative to gene flow and species-specific genomic properties, such as genomic architecture[1–4]. Local adaptation, i.e., the evolution of advantageous phenotypes in local selective environments, is inherently linked to these factors[5]. Divergent selection by local environmental conditions promotes local adaptation[6], while gene flow modifies the response to selection by modulating the distribution of the genes that underlie ecologically relevant traits[7]. Some gene flow may contribute to local adaptation through the supply of adaptive genetic variants, whereas high levels may prevent local adaptation by homogenising the gene pool[8]. Genomic architecture influences the response to selection by controlling recombination rates, inheritance and gene interactions[2].

Species in a common landscape may vary considerably in evolutionary potential, i.e., the capacity to adapt to ecological gradients and changing environments. This may have several consequences. First, it may lead to between-species variation in long-term population viability[9]. Second, it may induce variability among species in the occurrence and intensity of eco-evolutionary dynamics[10, 11]. For instance, species may differ in the relative contribution of ecological factors (e.g., resource limitation) and adaptive evolution to changes in population size. Evolutionary versatility of member species may in this way also impact community composition[12]. Quantifying variability in evolutionary potential as well as assessing shared and unique adaptive responses across species within heterogeneous landscapes may thus help to understand landscape-moderated biodiversity patterns[13]. A major step towards this goal is to identify to what extent selection, gene flow and genomic architecture contribute to variability in evolutionary potential among multiple species inhabiting the same landscape[14].

Here we assess whether species inhabiting the same landscape differ in pattern, spatial scale and environmental drivers of adaptive divergence. To do so, we perform a comparative analysis of phenotypic and genomic population divergence in two phylogenetically related fishes, the three-spined stickleback (Gasterosteus aculeatus Linnaeus, 1758) and the nine-spined stickleback (Pungitius pungitius (Linnaeus, 1758)), which diverged 13 million years ago[15]. The three-spined stickleback is a prime model for the study of adaptive evolution. It has evolved repeatedly in ecotypes that occur in a wide range of habitats from marine systems to rivers, lakes, ditches and ponds, and that cover all possible transitions from panmixia to complete and irreversible reproductive isolation[16]. Several studies have analysed the genetic basis of adaptation in the three-spined stickleback, focusing on single genes[17–19] to full genomes[20]. For instance, it has been shown that rapid adaptation can occur through selection on standing genetic variation, and is facilitated by strong chromosomal linkage of the genes involved[20]. Integration over various three-spined stickleback systems fuels the debate on the importance and genomic basis of parallel and non-parallel evolution[20–25].

The sound understanding of the mechanisms underlying adaptation and speciation makes the three-spined stickleback an excellent model to study how natural populations adapt to complex and heterogeneous landscapes. In turn, the nine-spined stickleback, which coexists with the three-spined stickleback across a wide range of coastal and inland waters, is excellent for the extension of this analysis towards multi-species communities, providing a more holistic view on the landscape processes shaping adaptive divergence. The distribution, ecology, behaviour, morphology, genetics and genomics of both species have been extensively compared[26–40]. Although both species are euryhaline, the evolutionary history of the three-spined stickleback is primarily bound to marine and coastal areas[41], while the nine-spined stickleback has mainly evolved in freshwater[42]. Other distinctions are that the nine-spined stickleback is a less manoeuvrable swimmer, has a stronger preference for closed, shallow waters and tolerates lower oxygen levels[30, 33, 43]. No formal comparison of the level of phenotypic and genomic divergence across major ecological gradients within the same landscape exists for both species. It therefore remains unknown to what extent they differ in evolutionary potential to deal with the challenges along the broad habitat gradient over which they coexist.

We investigate both species across a salinity gradient (brackish to freshwater) within exactly the same spatial matrix, located in the coastal lowlands of Belgium and the Netherlands (Fig. 1a). Four brackish water and four freshwater sites were selected among a set of ponds, streams and creeks (Fig. 1a and Supplementary Table 1). The dense network of water bodies in the area results in opportunities for moderate to strong gene flow across ecological gradients, maximising the opportunity to study the interaction between adaptive evolution, gene flow and genetic drift[5]. We start from the observation that population densities of the three-spined stickleback, but not of the nine-spined stickleback, are larger in freshwater than in brackish water (Supplementary Table 1). The three-spined stickleback thus dominates the freshwater sites, a pattern consistent over at least four seasons (Fig. 1b). Considering that adaptive evolution may have a measurable impact on population and community dynamics (see above), such a shift in population density of a species with marine ancestry possibly reflects the demographic signature of rapid freshwater adaptation. In order to characterise the degree of adaptive evolution, we first document and compare the strength of phenotypic and genomic signatures of adaptive divergence in each species. We focus on traits including body armour (responsive to predation and linked to the ionic environment[44]), body shape (responsive to hydrodynamics and linked to foraging behaviour[45]) and trophic morphology (responsive to prey availability). Genotyping by sequencing is used to identify genomic signatures of selection. Second, we study the degree of phenotypic and genomic parallelism to quantify to what extent environmental changes throughout the landscape affect the same traits, genes and gene functions. Finally, we evaluate the contribution of environmental vs. spatial factors to population divergence in both sticklebacks. This informs us to what extent geography (indicative for spatial isolation) and environment (indicative for divergent selection) covary in shaping phenotypic and genetic variation in the study species. The results provide insight in the mechanisms underlying differences in evolutionary versatility and ecological success among different yet related species inhabiting the same landscape.

## Results

**Signatures of adaptive divergence.** In order to assess the propensity for adaptation in the three-spined and the nine-spined stickleback, we compared both species for the magnitude of phenotypic divergence and genomic divergence at outlier (i.e., putatively adaptive) loci against a background of neutral (i.e., non-adaptive) population divergence. Clustering of the brackish water and the freshwater populations is identically based on the 12,684 and 10,068 neutral single-nucleotide polymorphisms (SNPs) in the three-spined and nine-spined stickleback, respectively (Fig. 1c, d). Both species also show a similar decline in genetic diversity with distance to the coast (Supplementary Fig. 1). Such highly congruent population genetic structure

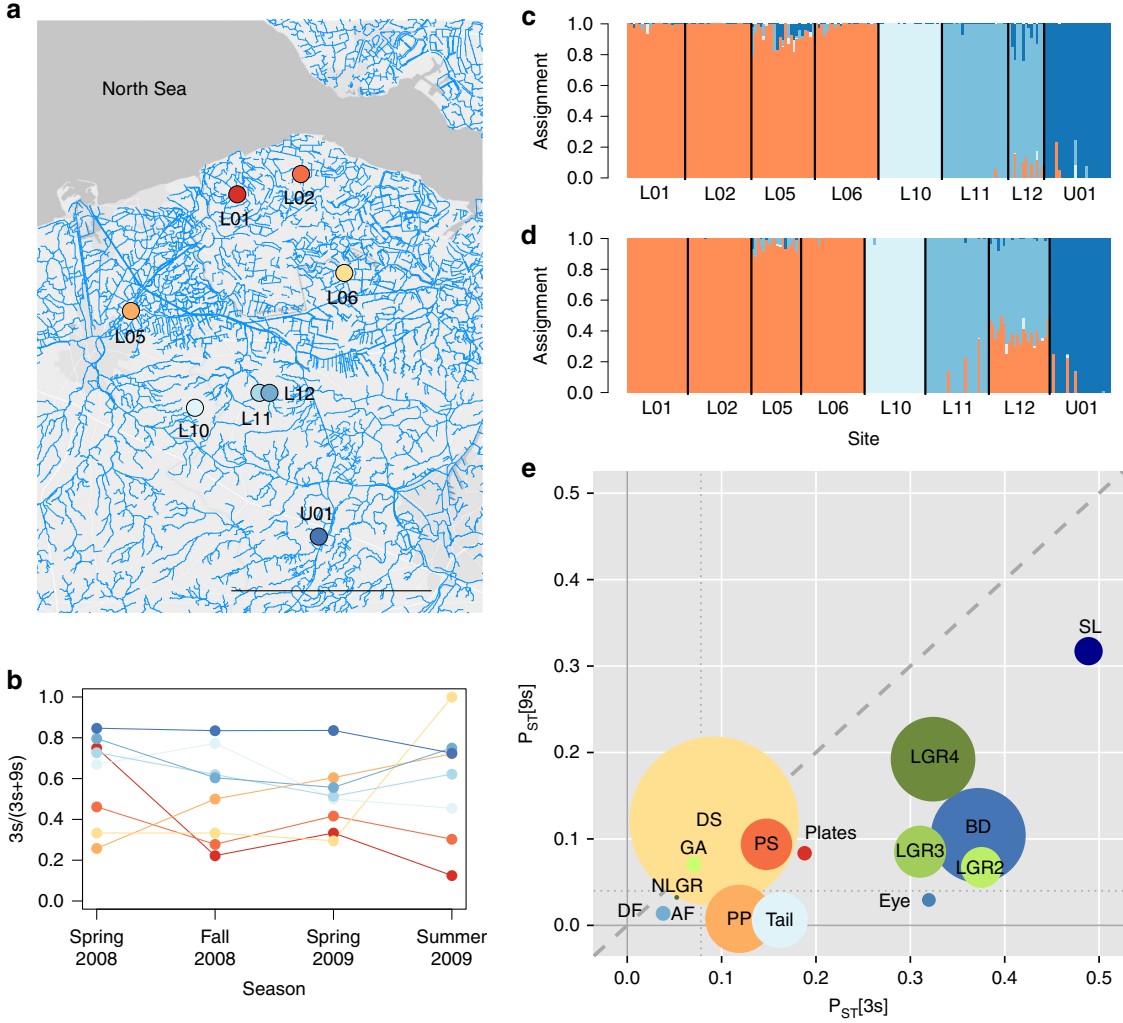

**Fig. 1** Locations and characteristics of eight coexisting populations of the three-spined (*3 s*) and nine-spined (*9 s*) stickleback. **a** Map of the Belgian-Dutch lowlands with brackish and freshwater sites represented by *orange-red* and *blue-shaded dots*, respectively. Site characteristics and sample size are listed in Supplementary Table 1. The length of the scale bar is 20 km. **b** Population density of the 3s stickleback relative to the total density of 3s and 9s stickleback, over four consecutive seasons. Across the eight sites, this proportion increased with decreasing salinity (Spearman $\rho = -0.71$; $P = 0.0465$). Sampling bias unlikely explains this result, because standard catches were done with a hand net at sites with little escape opportunity (see Supplementary Methods). **c–d** Bayesian analysis of neutral genetic structure among populations of the 3s stickleback **c** and 9s stickleback **d**, based on 12,684 and 10,068 neutral SNPs, respectively. We identified similar genetic clusters in both species, including a cluster corresponding to the brackish water populations (*orange*), and three clusters of freshwater populations corresponding to L10 (*light blue*), L11 and L12 (*blue*) and U01 (*dark blue*). **e** Phenotypic differentiation ($P_{ST}$) in the 3s vs. the 9s stickleback for 15 morphological traits, including standard length (*SL*) (for codes of other traits, see Supplementary Table 2). Shades of *red-orange*, *blue* and *green* represent armour traits, body shape and gill morphology, respectively. The size of the *circles* is indicative for the importance of parallel vs. non-parallel effects, quantified as the ratio of the corresponding effect sizes (see Supplementary Table 2). The *dashed line* represents the 1:1 line, indicating that $P_{ST}[3s]$ generally exceeds $P_{ST}[9s]$. The *dotted lines* mark the level of neutral genetic divergence (3s: $F_{ST} = 0.078$; *vertical line*; 9s: $F_{ST} = 0.040$; *horizontal line*)

suggests that the neutral genetic divergence among populations in the two species is shaped by similar evolutionary processes such as gene flow and genetic drift, contingent on the shared landscape. An overall lower genetic diversity in the three-spined stickleback suggests a stronger contribution of genetic drift in this species (Supplementary Fig. 1). Accordingly, effective population size ($N_e$) was generally lower in the three-spined stickleback than in the nine-spined stickleback, with particularly low values for the pond populations (L10 and L11; Supplementary Table 1).

Striking differences in the magnitude of population-level phenotypic divergence suggest a stronger response to environmental triggers in the three-spined stickleback than in the nine-spined stickleback. Effects of site (partial $\eta^2$) and levels of phenotypic differentiation ($P_{ST}$) for 14 morphological traits were larger in the three-spined stickleback than in the nine-spined stickleback (Table 1, Fig. 1e and Supplementary Tables 2 and 3). Likewise, $P_{ST}$ exceeded neutral $F_{ST}$ for seven traits in the three-spined stickleback, but only for one trait in the nine-spined stickleback (Table 2 and Supplementary Table 2), suggesting a stronger contribution of divergent selection or phenotypic plasticity in this species. A substantial proportion of the phenotypic variation in the two species could be attributed to differences between freshwater (L10-U01) and brackish water (L01-L05) populations (Supplementary Fig. 2). Specifically, in the three-spined stickleback, freshwater populations had fewer lateral plates, and shorter pelvic plates, anal fins, and gill rakers than the brackish water populations. In the nine-spined stickleback, freshwater populations had longer gill arches, shorter gill rakers

**Table 1 Single species and two species MANCOVA on 14 morphological traits in coexisting three-spined (3s) and nine-spined (9s) stickleback populations from eight sites**

| Species | Effect | Df | Wilk's $\lambda$ | F | P value | Partial $\eta^2$ (95 % CI) |
|---|---|---|---|---|---|---|
| 3s | Site | 7 | 0.02 | $F_{98,818.46} = 6.933$ | **<0.0001** | 0.421 (0.419–0.425) |
| | SL | 1 | 0.05 | $F_{14,128} = 164.366$ | **<0.0001** | 0.947 (0.946–0.948) |
| | Residuals | 141 | | | | |
| 9s | Site | 7 | 0.12 | $F_{98,666.67} = 2.706$ | **<0.0001** | 0.261 (0.259–0.265) |
| | SL | 1 | 0.03 | $F_{14,104} = 244.86$ | **<0.0001** | 0.971 (0.970–0.971) |
| | Residuals | 117 | | | | |
| 3s and 9s | Site | 7 | 0.11 | $F_{98,1558.4} = 6.79$ | **<0.0001** | 0.275 (0.273–0.277) |
| | Species | 1 | 0.02 | $F_{14,245} = 806.4$ | **<0.0001** | 0.979 (0.979–0.979) |
| | SL | 1 | 0.04 | $F_{14,245} = 387.15$ | **<0.0001** | 0.957 (0.956–0.957) |
| | Site by species | 7 | 0.27 | $F_{98,1558.4} = 3.68$ | **<0.0001** | 0.171 (0.170–0.173) |
| | Species by SL | 1 | 0.72 | $F_{14,245} = 6.65$ | **<0.0001** | 0.275 (0.272–0.280) |
| | Residuals | 258 | | | | |

Single species models test for differences between sites in each species. The two species model tests for phenotypic parallelism (effect of site) and non-parallelism (effect of site by species). Standard length (SL) is included in the models to correct for body size differences between individuals. Partial $\eta^2$ quantifies effect size. Significant P values are in bold. Models and error plots for single traits are provided in Supplementary Table 2 and Supplementary Fig. 2, respectively. Models for separate trait categories (armour, body shape and gill morphology) are provided in Supplementary Table 3

**Table 2 Phenotypic ($P_{ST}$) and neutral genetic ($F_{ST}$) differentiation in the three-spined stickleback (3s) and in the nine-spined stickleback (9s)**

| Species | Mean $P_{ST}$ (95 % CI) | Neutral $F_{ST}$ (95 % CI) | $P_{ST}/F_{ST}$ | $P_{ST} > F_{ST}$ |
|---|---|---|---|---|
| 3s | 0.19 (0.12–0.25) | 0.078 (0.076–0.080) | 2.38 | 7/14 (50 %) |
| 9s | 0.07 (0.04–0.09) | 0.040 (0.039–0.041) | 1.75 | 1/14 (7 %) |

Mean $P_{ST}$ represents the average $P_{ST}$ of 14 morphological traits (Supplementary Table 2; excluding body size). Neutral $F_{ST}$ was calculated based on 12,684 and 10,068 neutral SNPs in the 3s and 9s stickleback, respectively. The ratio of mean $P_{ST}$ over neutral $F_{ST}$ ($P_{ST}/F_{ST}$) and the proportion of single-trait $P_{ST}$ values that significantly exceeded neutral $F_{ST}$ ($P_{ST} > F_{ST}$) are also shown

and smaller eyes than the brackish water populations (Supplementary Fig. 2).

We then tested whether a similar trend could also be observed at the genomic level. One indication for genomic adaptation is provided by the proportion of outlier loci, i.e., loci that show weaker or stronger genetic differentiation than expected based on the overall (genome-wide) level of differentiation[46]. Based on three different methods (i.e., LOSITAN, ARLEQUIN and BAYESCAN), we found that the proportion of positive outliers (i.e., potential targets of directional selection) was 2.5–10 times larger in the three-spined stickleback (66–283 outlier loci; 0.52–2.22 %) than in the nine-spined stickleback (8–22 outlier loci; 0.08–0.22 %; Fig. 2 and Supplementary Table 4). SNP typing in the nine-spined stickleback species was not performed with the help of the three-spined stickleback genome, thereby avoiding a bias towards homologous genomic regions (note that no reference genome (RG) is currently available for the nine-spined stickleback). The estimated proportions of outlier loci are therefore directly comparable between the species. Gene ontology (GO) analysis of the outlier loci (including genomic regions 5 kb upstream and downstream from these loci) revealed unique biological processes and molecular functions for the three-spined stickleback, but not for the nine-spined stickleback (Supplementary Table 5). Arguably, the low number of outlier loci in the latter may render this comparison redundant.

In summary, we observed stronger phenotypic differentiation and a stronger genomic signature of adaptive divergence in the three-spined stickleback than in the nine-spined stickleback. This suggests a stronger propensity for adaptation in the former, which could be rooted in genetic properties, phenotypic plasticity or a combination of both[32, 47, 48]. Genetic drift was also more pronounced in the three-spined stickleback, but the strong levels of phenotypic divergence and the presence of outlier loci suggests that neutral divergence does not overwhelm adaptation. Based on populations from exactly the same spatial matrix, it is unlikely

that this outcome is strongly influenced by extrinsic factors (e.g., selective environments and degree of spatial isolation) other than those that directly interact with species-specific properties such as life history and genomic architecture (see below). Importantly, the absence of a strong signature of adaptive divergence in the nine-spined stickleback does not imply that the populations of this species are not adapted, since they might already be pre-adapted to the ecological gradients in the landscape. However, the aforementioned numerical advantage of the three-spined over the nine-spined stickleback in freshwater (Fig. 1b) suggests a remarkable ecological success that could possibly be attributed to evolutionary versatility. Indeed, it is known that freshwater adaptation in the three-spined stickleback can occur rapidly[49–51]; the associated fitness increase may enhance population growth[52]. Whether or not such increase is fast enough to impact community composition depends on the effect of ecological factors on the two species (e.g., resource limitation), and may be influenced by species interactions such as competition[10, 14].

**Parallel and non-parallel population divergence.** Examples of parallel evolution suggest that organisms experiencing similar selective pressures can develop similar responses to cope with those pressures[24, 53–56]. In such instances, natural selection drives phenotypic change in a repeatable and seemingly deterministic manner, even in lineages that have been separated for millions of years[57]. Yet, these patterns do not appear in other taxa, questioning the general importance of such deterministic selective forces, how they are modified by gene flow and how they are modulated by species-specific genomic properties[25, 58].

In order to assess the degree of parallel vs. non-parallel phenotypic divergence between both stickleback species, we quantified the effect of site (indicative of phenotypic parallelism) as well as the site by species interaction term (indicative for non-parallelism) on phenotypic variation. Both components explained

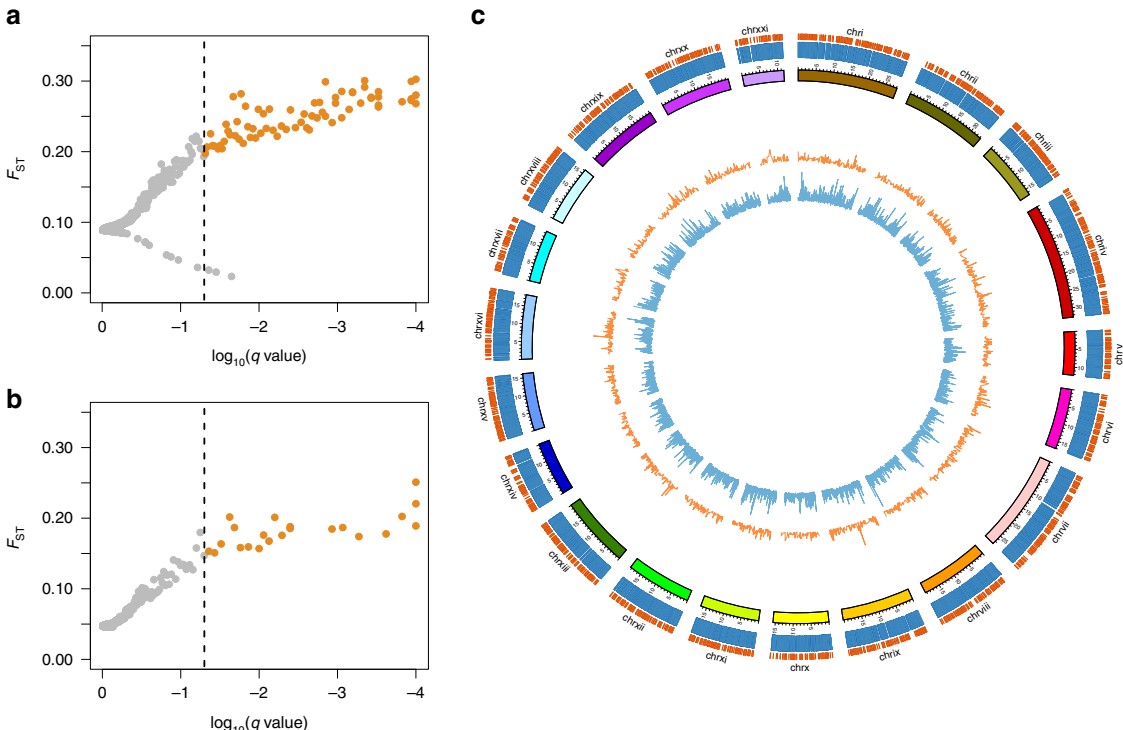

**Fig. 2** Genomic population divergence in eight coexisting populations of the three-spined and nine-spined stickleback. BAYESCAN outlier detection in data sets containing **a** 12,754 SNPs for the three-spined and **b** 10,090 SNPs for the nine-spined stickleback. Genetic divergence ($F_{ST}$) of each locus is plotted against the log-transformed $q$ value. Loci in *orange* are classified as candidates under divergent selection. Loci in *grey* are classified as neutral or are candidates under balancing selection. **c** Map of the three-spined stickleback genome. The outermost (*orange*) and second outermost (*blue*) bars represent the mapped genotyped loci in the nine-spined and the three-spined stickleback, respectively. The *coloured blocks* represent the different linkage groups. The *orange* (nine-spined stickleback) and *blue* (three-spined stickleback) *inner circles* represent line charts showing $F_{ST}$ for all SNPs throughout the genome

significant variation for the 14 morphological traits considered, but the effect size of parallelism was significantly larger (1.61 times) than the effect of non-parallelism (Table 1). The average correlation between the three-spined and nine-spined stickleback population means for these traits was moderate (Pearson $R = 0.35$; 95% CI: 0.13–0.58), but significance was observed for 5 out of 14 traits (36%; Supplementary Fig. 3). Specifically, populations of both species showed similar changes in dorsal spine length, body depth and gill raker length. Across traits, the ratio of parallel vs. non-parallel effect sizes ($R^2$ ratio; Supplementary Table 2) increased with phenotypic differentiation ($P_{ST}$) in the nine-spined stickleback (Pearson $R = 0.55$; $P = 0.0397$), while this relationship was weak in the three-spined stickleback (Pearson $R = 0.22$; $P = 0.45$). This suggests that phenotypic parallelism between the two sticklebacks is constrained by weak phenotypic divergence in the nine-spined stickleback.

In contrast to the phenotypic level, there were no indications for parallelism between both species at the genomic level. Indeed, among the 933 genes that were annotated in both species (i.e., homologous genes), not a single one was flanked by a common outlier locus. In comparison, 78 out of 7127 annotated genes in the three-spined stickleback (1.09%) and 6 out of 2133 annotated genes in the nine-spined stickleback (0.28%) were flanked by an outlier locus.

In summary, both species showed substantial phenotypic parallelism in the absence of genomic parallelism, and despite substantial differences in the magnitude of phenotypic divergence. Hence, exposure to similar selection pressures among species inhabiting a common landscape might promote the evolution or development of similar phenotypes. In a functional genomic analysis of the architecture of skeletal elements,

Shapiro et al.[39] demonstrated that convergent evolution among the three-spined and nine-spined stickleback has different genetic origins. While the probability of gene reuse upon parallel and convergent evolution declines with divergence time between taxa[59], it is exciting to observe phenotypic parallelism among related but independent species in exactly the same landscape. It demonstrates that the evolution of similar phenotypes to the same selective environments might primarily involve different genes.

**Spatial and environmental drivers of population divergence.** Both natural selection and neutral processes, such as gene flow and drift, contribute to population divergence[3, 8]. Together, these processes determine the distribution of ecologically relevant genes, and therefore influence the capacity of natural populations to adapt to local selective environments. We therefore extend our comparative approach by simultaneously assessing the contribution of environmental drivers (indicative for divergent selection) and spatial drivers (indicative for spatial isolation) to phenotypic and genomic population divergence in both species. To do so, we performed multivariate redundancy analyses (RDA) to attribute explainable variation in phenotypic traits, neutral SNPs and outlier SNPs to space, environment or their joint effect. This analysis integrates two classical models of population structure, isolation-by-distance and isolation-by-adaptation[60, 61]. These models assume that population structure is shaped independently by geographical distance and ecological divergence, respectively, while here we also consider their joint effect (see also refs [62–64]).

The proportion of variation explained (PVE) in phenotype and genotype by spatial and environmental variables was always larger in the three-spined stickleback than in the nine-spined

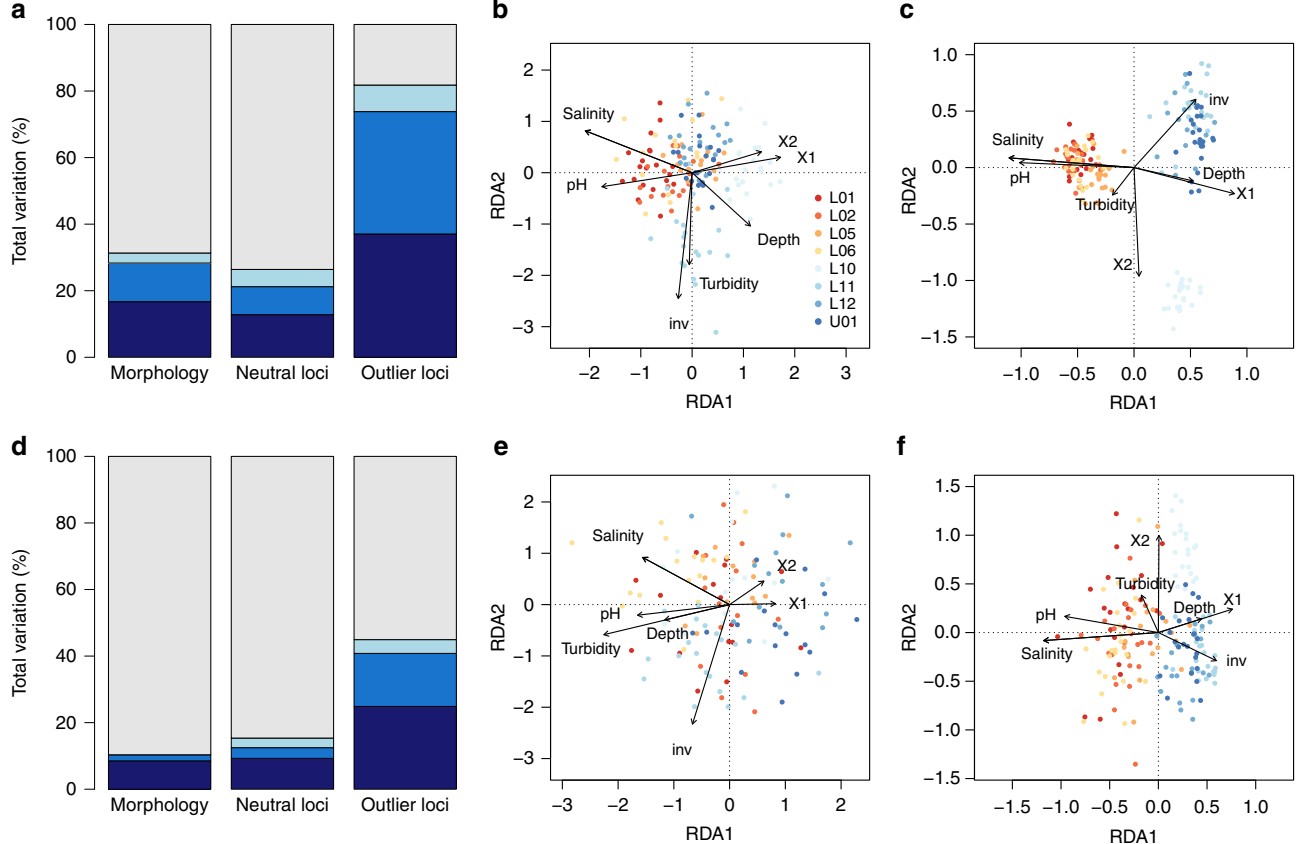

**Fig. 3** Redundancy analysis of population divergence in two coexisting stickleback species. *Barplots* represent the proportion of the total variation in morphology, neutral loci and outlier loci that can be explained by space (*light blue*), environment (*dark blue*) or their interaction (*blue*). *Scatterplots* show the first (RDA1) vs. the second (RDA2) dimension. Individuals from brackish and freshwater sites are represented by *orange-red* and *blue-shaded dots*, respectively. Site characteristics and sample size are listed in Supplementary Table 1. **a** Variance partitioning of morphological and genomic data in the three-spined stickleback. **b** Morphological divergence vs. space and environment in the three-spined stickleback. **c** Genomic divergence at outlier loci vs. space and environment in the three-spined stickleback. **d** Variance partitioning of morphological and genomic data in the nine-spined stickleback. **e** Morphological divergence vs. space and environment in the nine-spined stickleback. **f** Genomic divergence at outlier loci vs. space and environment in the nine-spined stickleback

stickleback (Fig. 3a, d and Supplementary Tables 6 and 7). However, both species differed markedly in the relative contribution of space and environment. Particularly, in the nine-spined stickleback, environmental effects always accounted for the largest fraction of explainable variation in morphology, neutral loci and outlier loci (Fig. 3d). In the three-spined stickleback, a substantial fraction of the variation could be attributed to the joint effect of space and environment (Fig. 3a). The difference between the species for this joint effect was stronger for morphology (PVE: 12% in the three-spined stickleback vs. 2% in the nine-spined stickleback) and outlier loci (PVE: 37% in the three-spined stickleback vs. 16% in the nine-spined stickleback) than for neutral loci (PVE: 8.4% in the three-spined stickleback vs. 3.2% in the nine-spined stickleback). Overall, this suggests that population divergence for traits and outlier loci is facilitated mainly by environmental factors in the nine-spined stickleback, but by the covariance between environmental and spatial factors in the three-spined stickleback.

Forward selection analyses identified salinity as the most important determinant of phenotypic and putatively adaptive genetic variation in the three-spined stickleback (Fig. 3b, c and Supplementary Table 6). In the nine-spined stickleback, salinity was the most important determinant of putatively adaptive genetic variation, while turbidity and the density of macro-invertebrate predators were the strongest determinants of

phenotypic variation (Fig. 3e, f and Supplementary Table 7). Latent factor mixed models revealed that environmental variables and outlier loci were more often correlated in the three-spined stickleback than in the nine-spined stickleback (Supplementary Fig. 4). Salinity, closely followed by pH, correlated with the largest percentage of outlier loci in both species (Supplementary Fig. 4).

In summary, population divergence for traits and outlier loci in the nine-spined stickleback was predominantly shaped by environmental variation, and involved variables with a clinal (salinity) as well as a patchy distribution (turbidity and the density of macro-invertebrate predators). In contrast, in the three-spined stickleback, population divergence for traits and outlier loci was strongly shaped by the joint effect of space and environment, and primarily involved salinity. This may imply that divergent selection and spatial isolation in this species are strongly intertwined, and truly coincide along the brackish water freshwater gradient.

## Discussion

Differences in population divergence among species within landscapes may be rooted in various species-specific properties that may influence their propensity for adaptation. Here we discuss three major factors that may contribute to such differences in both species of stickleback: genomic architecture, gene flow and life history.

The propensity for adaptation in the three-spined stickleback has been attributed to several genomic features[20, 23, 65–68]. For instance, the arrangement of freshwater-adapted alleles in marine populations in linkage blocks[20, 66, 69] has been suggested to facilitate the response to selection upon freshwater colonisation[70]. In particular, Linkage Group IV, which includes the Ectodyplasin (*EDA*) gene, the major locus for lateral plate variation, contains an extended region of linkage disequilibrium and the entire chromosome may be selected as a unit upon freshwater colonisation[69]. However, for the three-spined stickleback populations in this study, there are strong indications that gene flow may in fact counter the effect of selection on this locus[21, 71]. Furthermore, a number of inversion polymorphisms have been associated with genomic divergence between three-spined stickleback ecotypes[20, 68], and may promote adaptation in the face of strong gene flow[2]. While such inversions may at least partially explain the high $F_{ST}$ values for the three-spined stickleback in our study system, a higher marker density would be required to detect them. Insight into the genomic synteny of the three-spined and the nine-spined stickleback is steadily growing[28, 29, 34, 38]. However, which genomic features influence population divergence in the nine-spined stickleback remains largely unknown. The genomic basis of differences in evolutionary versatility between both species therefore remains to be identified. In general, a pattern of distinct ecoresponsive genomic regions in the two species is emerging (see Discussion section in ref. [27]). Further analysis of structure and biological function of target genomic regions will be required to evaluate the effect size of the observed discrepancy between the two species in terms of outlier loci. Importantly, by studying both sticklebacks in exactly the same spatial matrix, we here document non-parallel genomic signatures of adaptation even when species effectively experience the same selective environments.

Gene flow might influence the propensity for adaptation by modulating the distribution of the genes that underlie ecologically relevant traits[7]. The overall lower genetic diversity in the three-spined stickleback suggests that this species is experiencing less gene flow and more genetic drift than the nine-spined stickleback (Supplementary Fig. 1). Strong adaptive divergence is compatible with such low levels of gene flow under a pure isolation-by-adaptation mechanism. Under this model, and as opposed to the isolation-by-distance model, ecological divergence constrains neutral genetic variation more strongly than spatial isolation[60]. However, both sticklebacks mainly differed in how strong spatial and environmental factors are intertwined in explaining population divergence—arguing for the use of models that assess both effects simultaneously[64]. While differences in gene flow per se are thus probably no main contributor to differences in adaptive divergence in both species, the effects of gene flow on adaptive processes may still depend on the geographical context. For instance, in the case of the three-spined stickleback, gene flow from marine or anadromous populations from outside our study area may affect adaptation in the brackish water and freshwater populations[21]. However, details on the existence of a distinct marine population as well as on the importance of anadromy in the region are currently lacking, and hence such effects remain uncertain.

Finally, differences in life history might influence adaptive processes when different levels of phenotypic plasticity lead to a discrepancy in the strength of natural selection between species[5]. For instance, the nine-spined stickleback is more resilient to extreme temperatures in summer owing to a better tolerance of hypoxic conditions[43]. In contrast, the three-spined stickleback may not tolerate such conditions, which may lead to stronger extinction–recolonisation dynamics and population turnover, possibly associated with selective sweeps. Indeed, in the course of this study, we have regularly observed local extinction and recolonsiation of three-spined stickleback populations, in particular at the freshwater sites. This may lead to a less stable population structure and could explain the overall smaller effective population size at these sites. Extinction–recolonisation dynamics are generally unfavourable for local adaptation[5]. Yet, in the case of the three-spined stickleback, they likely occur along the brackish water freshwater gradient, where gene flow might initially be fuelling freshwater adaptation. This may provide an additional explanation of why environmental and spatial factors are strongly intertwined in shaping population divergence in the three-spined stickleback.

In conclusion, this study revealed three important aspects of how members of a community diverge in a shared landscape. First, while species may show a highly concordant spatial genetic structure, they may strongly differ in their responses to environmental contrasts. Second, parallel effects may exceed non-parallel phenotypic responses to these environmental contrasts, but such effects are not necessarily reflected at the genomic level. Third, species may differ in the contribution of spatial, environmental and joint effects to population divergence. Together, these aspects reflect different ways for persistence in the same landscape, which may represent a key element underlying differences in ecological resilience between species. In species such as the three-spined stickleback, a genome wired for rapid adaptation and a strong dispersal capacity fuelling gene flow may lead to strong adaptive divergence at short spatiotemporal scales. In other species, such as the nine-spined stickleback, weaker divergence may be the result of stronger tolerance for harsh local environmental conditions. The results of this study facilitate the understanding of variation in evolutionary versatility and ecological success across species inhabiting heterogeneous environments. These and other insights generated by multi-taxa genomic approaches over large and ecologically diverse landscapes are key to understand landscape-moderated biodiversity patterns, and are therefore becoming increasingly important for the study of evolving metacommunities (e.g., landscape community genomics), nature conservation planning and landscape management[14, 72].

## Methods

**Study area and species**. The coastal lowlands of Belgium and the Netherlands harbour brackish and freshwater habitats of Holocene origin with variable connectivity to adjacent estuaries and the open sea[40, 73]. The three-spined and nine-spined stickleback dominate the local fish communities. Postglacial expansion by marine populations and subsequent freshwater colonisations characterise the phylogeographic history of the three-spined stickleback[41]. The phylogeography of the nine-spind stickleback has mainly progressed in freshwater[42]. More recently, the distribution of both species in our study area has been influenced by a shifting coastline after the last glacial, and by the construction of dikes and drainage systems (see Supplementary Methods for further details).

**Field work**. Field sampling was done in accordance to European directive 2010/63/EU and explicit permission of the Agency for Nature and Forests. Four brackish sites and four freshwater sites were visited seasonally between the spring of 2008 and the summer of 2009 to obtain habitat characteristics, and estimates of population density (Fig. 1 and Supplementary Table 1). Field work was performed as described in ref. [21] (see Supplementary Methods for further details). A minimum of 24 adult individuals per site and species, all obtained in the spring of 2009, were selected for subsequent morphological and genomic characterisation. Final sample sizes (i.e., excluding individuals with missing data or low read number) are listed in Supplementary Table 1.

**Morphological characterisation and analyses**. We scored 15 morphological traits in both species, including standard length, four armour traits, five body shape traits and five gill traits (see Supplementary Table 2). MANOVAs and ANOVAs were used to partition the phenotypic variation into parallel (effect of site), species-specific (effect of species) and non-parallel (effect of site-by-species interaction) components (see Supplementary Methods for further details). To explicitly test which phenotypic traits differ between populations from freshwater and brackish

water habitat, we also performed ANOVAs with site nested in habitat type. In order to compare the level of phenotypic differentiation directly with the level of genetic differentiation in each species, we calculated $P_{ST}$, an index which quantifies the proportion of among-population phenotypic variance in quantitative traits[48]. $P_{ST}$ values along with 95% Bayesian confidence intervals were estimated following Leinonen et al.[47] (see Supplementary Methods for further details).

**Genomic characterisation and analyses.** For 192 individuals of each species, SNPs were generated using genotyping-by-sequencing[74] on an Illumina HiSeq 2000 sequencing platform. Currently, a RG is available for the three-spined stickleback, but not for the nine-spined stickleback. We therefore performed SNP typing in two ways. First, we applied de novo-based SNP typing in both species. Second, we performed SNP typing using the three-spined stickleback genome as a RG for both species. For the three-spined stickleback, the use of the RG resulted in more SNPs (RG: 12,754 SNPs; de novo: 4760 SNPs), but both strategies resulted in very similar outcomes for all downstream analyses. We therefore only present the results of the RG-based SNP typing. For the nine-spined stickleback, the use of the three-spined stickleback RG resulted in less SNPs (RG: 3877 SNPs; de novo: 10,090 SNPs)—a result which can be explained by low mapping success. Given this lower number, and given that the RG-based SNP typing limits the detection of SNPs to homologous regions (which may bias some of the genomic characteristics that were of primary interest for the comparison between both species), we only present the de novo-based SNP data set for this species (see Supplementary Methods for further details). The analyses of genomic differentiation included an assessment of neutral genetic diversity (expected heterozygosity; $H_e$), neutral genetic structure (standardised allelic variance; $F_{ST}$), effective population size ($N_e$), the identification of genomic signatures of selection with the software packages LOSITAN[75], ARLEQUIN[76] and BAYESCAN v2.01[77], and an association analysis between environmental variables and SNPs using the software package LFMM[78]. For both species, SNPs were mapped against the three-spined stickleback genome, and single-SNP $F_{ST}$ values were visualised using Circos plots. Finally, GO terms were determined for the genes 5 kb upstream and downstream of all outlier loci (as identified with BAYESCAN). Note that the visualisation with Circos plots and the identification of genes depend on the mapping success to the three-spined stickleback genome, which obviously differs between both species.

**Variance partitioning.** For each species, RDA[79] were conducted to partition the explainable phenotypic variation, neutral allelic variation and allelic variation at outlier loci (as identified with BAYESCAN) into those attributable to spatial factors (SPACE), environmental factors (ENV) and their combined effect (ENV + SPACE). The full, partial and joint contributions of SPACE and ENV to the explainable phenotypic or genetic variation were estimated and tested for significance, and the most influential single explanatory variables were identified (see Supplementary Methods for further details).

**Data availability.** SNP-based multilocus genotypes (VCF files) and morphological, spatial and environmental data used in analyses are archived at the Dryad Digital Repository (doi:10.5061/dryad.8sm32). All other data is included in the article and its supplementary information files.

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

## Acknowledgements

We thank Nele Boon, Dorien Daneels, Camille De Raedemaeker, Jo-Ann De Roos, Sarah Geldof, Lize Jacquemin, Nellie Konijnendijk, Koen Martens, Sanne Ruyts, Frank Spikmans, Kathelijne Szekér, Nicolas Thiercelin, Sarah Tilkin, Tina Van den Meersche, Dorien Verheyen and Inne Withouck for field support and technical assistance, and Annelies Bronckaers, Daniel Berner, Federico Calboli, Karl Cottenie, Christophe Eizaguirre, Andrew Hendry, Martin Kalbe, Tuomas Leinonen, Pieter Lemmens, Zuzana Musilová and Asbjørn Vøllestad for insightful comments. Research was sponsored by the Research Foundation – Flanders (Grant 1526712N to J.A.M.R.), the University of Leuven (KU Leuven Centre of Excellence PF/10/07) and BELSPO-Interuniversity Attraction Poles (Project P7/04). J.A.M.R. received an EU Marie Skłodowska-Curie Fellowship (IEF 300256).

## Author contributions

J.A.M.R. and A.C. conceived the study and performed the bioinformatic analyses. J.A.M.R., P.I.H., I.V., B.H. and G.E.M. collected the data. J.A.M.R., A.C., P.I.H., G.E.M, L.D.M. and F.A.M.V. interpreted the results and wrote the manuscript.

## Additional information

**Competing interests:** The authors declare no competing financial interests.

