## [Peer Review File · Nature Communications]

Reviewers' comments:

Reviewer #1 (Remarks to the Author):

In this manuscript the authors have collected a large amount of environmental, phenotypic and genotypic data on stickleback fish in order to understand how organisms adapt to common environmental challenges. Overall I find this paper very interesting and well executed. The methods used and analyses performed seem appropriate and state-of-the-art. I would like to congratulate the authors with their effort.

I only have one "major" comment, and a few minor comments.

The title of the manuscript is "Adaptive divergence in a common landscape", and the use of words like adaptation and adaptive is used extensively. The authors use a standard definition of local adaptation: "the evolution of advantageous phenotypes in local selective environments". It is inherently difficult to study if a phenotype is advantageous in a give environment (more fit than other phenotypes) based on field studies alone. Usually some experimental testing is needed. I thus think the authors needs to be more careful and discuss if they indeed document adaptation. I think they do, but their wording may at places be too strong. In relation to this, I also think they under-communicate to what degree phenotypic plasticity may explain/impact on the phenotypic variation that they observe. There is a mention of plasticity on L273, and then in the context of adaptive plasticity. Plasticity is not always adaptive; so I think the authors should be more careful.

Minor

Study locations (Table S1): add some variance metric to the table. I expect large variability in salinity, pH and turbidity in brackish water. Further, the range of salinity observed is very small (2.35 – 0.18 psu); how do you define fresh/brackish water? Also, the density of 3s and 9s individuals is very variable, probably depending on timing of recruitment of the two species. I would add some information on this. In L93 you state that 3s dominates in freshwater – I find the word dominate to strong given the variability.

On L278-9 you mention local extinction and recolonization. Did this happen at your selected locations during the study period? Which sites? Can you estimate N_e and potential bottleneck effects based on your data to inform on this?

Fig 3. In the legend you refer to Table 1 – should probably be Table S1.

Asbjørn Vøllestad

Reviewer #2 (Remarks to the Author):

This is a nicely written comparative genomic (and phenotypic) study of two species of sticklebacks where authors have compared the degree and similarity of genetic and

phenotypic differentiation across multiple sites occupied by both species. The underlying data is sound, and analyses appear to be (mostly) well conducted, and the results are generally interesting. In my opinion, this is a solid and interesting study, with one major caveat that can be dealt with careful re-wording of parts of the manuscript.

The caveat to which I am referring to is the authors' decision to interpret the observed patterns as adaptive. In my opinion, this too strong (cf. the title) and unnecessary. I believe the results are interesting even if the adaptive interpretations are tuned down a bit. For the first, a lot of thrust is put on interpreting the phenotypic patterns as adaptive (e.g. L24-25; more examples below) in spite of the fact the genetic basis for these phenotypic divergences are not know. In fact, all the parallelism (or non parallelism) could be equally parsimoniously explained by shared environmental effects (i.e. non-genetic environmental induction). In short, without common garden experiments, it is logically impossible to deduce whether the observed patterns reflect adaptation. This kind of interpretations constitute a common problem in current evolutionary biology literature, and pleas for community to steer away from this has been repeatedly voiced (for a recent one see: *Evolutionary Applications* 7:1-14).

The interpretations of the phenotypic data aside, evidence for adaptive differentiation from the genomic data is logically more easily made. However, here too, one should keep in mind the inference from the outliers is prone to false positives. Hence, it is prudent consider outlier information as suggestive of adaptation, rather than claim it to be hard evidence. I think the authors have made good job in this most of the time, but I was left to wonder why more recent procedures such as OutFLANK, which have been shown to reduce detection false positives, was not used?

Some detailed remarks:

L60. "phenotypic and genomic adaptive divergence". See my comments above.

L96-97. Ditto.

L120-121. Ditto.

L125. Interestingly, here plasticity is mentioned, but even if so, the reference is to "adaptive plasticity". Why would it need to be adaptive? I do not think you have way of inferring this from your data.

L144. "phenotypic and genomic adaptive divergence". See my comments above – this simply unjustified: you are stretching your interpretations beyond the data.

L219. "adaptive divergence is..." see above.

L254-256. What about: Rastas P, et al 2016. *Genome Biology & Evolution* 8:78-93?

L286. "Adapt". Would respond be more prudent?

L346. From supplementary methods I found that you used also Arlequin. This came bit as surprise as it was not mentioned here. Why?

Table 2. I would be more satisfactory to see formal analyses whether PST estimates are in fact significantly higher than FST. It is not straightforward to deduce if this the case from the numbers given in this table.

Reviewers' comments:

Reviewer #1 (Remarks to the Author):

In this manuscript the authors have collected a large amount of environmental, phenotypic and genotypic data on stickleback fish in order to understand how organisms adapt to common environmental challenges. Overall I find this paper very interesting and well executed. The methods used and analyses performed seem appropriate and state-of-the-art. I would like to congratulate the authors with their effort.

I only have one “major” comment, and a few minor comments.

The title of the manuscript is “Adaptive divergence in a common landscape”, and the use of words like adaptation and adaptive is used extensively. The authors use a standard definition of local adaptation: “the evolution of advantageous phenotypes in local selective environments”. It is inherently difficult to study if a phenotype is advantageous in a given environment (more fit than other phenotypes) based on field studies alone. Usually some experimental testing is needed. I thus think the authors needs to be more careful and discuss if they indeed document adaptation. I think they do, but their wording may at places be too strong. In relation to this, I also think they under-communicate to what degree phenotypic plasticity may explain/impact on the phenotypic variation that they observe. There is a mention of plasticity on L273, and then in the context of adaptive plasticity. Plasticity is not always adaptive; so I think the authors should be more careful.

A1 – We agree and have edited the manuscript throughout to attenuate our wording and tone down the claim that we have documented adaptation, in particular because the observed patterns might to some degree reflect non-adaptive or neutral divergence. This is reflected in the new title, while in the abstract we mention the interplay with genetic drift. For the three-spined stickleback we consistently find patterns of population divergence that are hard to explained by neutral divergence alone. Here, we still argue that adaptive processes must be important (at least more important than in the nine-spined stickleback), but we also explicitly discuss the contribution of neutral processes (e.g. **Results > Signatures of adaptive divergence > final paragraph**). Furthermore, we mention that both genetic properties as well as phenotypic plasticity may be involved. We do not speculate to what degree, since our genomic data do not include known QTLs that would allow to directly compare genes and traits in both species. Finally, in the discussion we point out that neutral and adaptive processes may truly coincide in driving population divergence.

Minor

Study locations (Table S1): add some variance metric to the table. I expect large variability in salinity, pH and turbidity in brackish water. Further, the range of salinity observed is very small (2.35 – 0.18 psu); how do you define fresh/brackish water? Also, the density of 3s and 9s individuals is very variable, probably depending on timing of recruitment of the two species. I would add some information on this. In L93 you state that 3s dominates in freshwater – I find the word dominate to strong given the variability.

A2 - We have added standard deviations to **Supplementary Table 1** where applicable. The salinity data in psu have been converted from conductivity measurements in microS/cm. Sites with measurements that had consistently low salinities (i.e. equivalent to conductivities < 1000 microS/cm) were classified as freshwater sites. The salinity at the brackish water sites was on average higher than 1000 microS/cm, and was indeed more variable owing to the irregular influx of sea water. We have added this information to **Supplementary Methods > field work**. For pH and turbidity this was not observed, probably due to terrestrial effects.

Furthermore, the dominance of the three-spined stickleback in freshwater is indeed not absolute, but it is statistically supported through a significant negative correlation of the relative proportion of three-spined stickleback with salinity (see **Figure legend 1**), as well as through a significant habitat effect in a general linear model for the data of Figure 1B:

```
> lm3rand <- lme(relprop3s ~ season+habitat, data = OUT0, random = ~ 1 | Site)
> Anova(lm3rand,type=3)
Analysis of Deviance Table (Type III tests)
Response: relprop3s

            Chisq Df    Pr(>Chisq)
Intercept    40.8559  1  1.639e-10 ***
Season        1.5471  3  0.6714442
Habitat       12.9275  1  0.0003238 ***
```

We therefore believe that it is statistically and ecologically appropriate to say that the three-spined stickleback was generally the dominant species at the freshwater sites.

On L278-9 you mention local extinction and recolonization. Did this happen at your selected locations during the study period? Which sites? Can you estimate N_e and potential bottleneck effects based on your data to inform on this?

A3 – Local extinctions and strong reductions in population size occurred during summer droughts at all four freshwater sites (L10-U01). This was observed for the first time in the summer of 2010, i.e. after the data for this study were collected (spring 2008-summer 2009), but similar droughts likely occurred prior to this period. We also observed that the nine-spined stickleback sometimes manages to survive these droughts in very shallow water. So, the stochastic genetic effects should be more severe for the three-spined stickleback, and likely represent sequels of reductions in population size and founder effects. We have now added N_e calculations (LD method). They reveal an overall lower N_e for the three-spined stickleback than for the nine-spined stickleback, with particularly low values for L10 and L11, which happen to be the two pond populations. Methods (See **methods > Genomic characterisation and analyses** and **Supplementary Methods > Data Analysis > Neutral Genetic Structure**) and results (**Results > Signatures of adaptive divergence** and **Discussion > paragraph 4**) have been added. We did not run bottleneck tests, since as mentioned above, the scenario of contemporary extinction and recolonization might not reflect pure bottlenecks but also founder effects.

Fig 3. In the legend you refer to Table 1 – should probably be Table S1.

A4 – Indeed, we have changed this into “Supplementary Table 1”.

Asbjørn Vøllestad

Reviewer #2 (Remarks to the Author):

This is a nicely written comparative genomic (and phenotypic) study of two species of sticklebacks where authors have compared the degree and similarity of genetic and phenotypic differentiation across multiple sites occupied by both species. The underlying data is sound, and analyses appear to be (mostly) well conducted, and the results are generally interesting. In my opinion, this is a solid and interesting study, with one major caveat that can be dealt with careful re-wording of parts of the manuscript.

The caveat to which I am referring to is the authors' decision to interpret the observed patterns as adaptive. In my opinion, this too strong (cf. the title) and unnecessary. I believe the results are interesting even if the adaptive interpretations are tuned down a bit. For the first, a lot of thrust is put on interpreting the phenotypic patterns as adaptive (e.g. L24-25; more examples below) in spite of the fact the genetic basis for these phenotypic divergences are not know. In fact, all the parallelism (or non parallelism) could be equally parsimoniously explained by shared environmental effects (i.e. non-genetic environmental induction). In short, without common garden experiments, it is logically impossible to deduce whether the observed patterns reflect adaptation. This kind of interpretations

constitute a common problem in current evolutionary biology literature, and pleas for community to steer away from this has been repeatedly voiced (for a recent one see: Evolutionary Applications 7:1-14).

A5 – We agree, see answer A1 for the changes we have made in this respect.

The interpretations of the phenotypic data aside, evidence for adaptive differentiation from the genomic data is logically more easily made. However, here too, one should keep in mind the inference from the outliers is prone to false positives. Hence, it is prudent consider outlier information as suggestive of adaptation, rather than claim it to be hard evidence. I think the authors have made good job in this most of the time, but I was left to wonder why more recent procedures such as OutFLANK, which have been shown to reduce detection false positives, was not used?

A6 – OutFLANK has been developed to strongly reduce the detection of false positives as compared to other methods, in particular by taking into account that some populations might be more closely related than others (Whitlock & Lotterhos 2015). We have now tested this approach, and it is indeed much more conservative since neither for the three-spined stickleback, nor for the nine-spined stickleback, the software identified significant outliers under the default parameter values. Unless this indicates that OutFLANK is too conservative or lacks power, these results imply that there are no genes that strongly contribute to local adaptation in both species. While this might be true (e.g. detectable selection at the *Eda* gene in the three-spined stickleback seems to be overruled by gene flow; Raeymaekers et al 2014), we are somewhat skeptical about this result given that loci in the three-spined stickleback with F_{st} values exceeding 0.8 were still classified by OutFLANK as neutral loci. In fact, by performing one of the recommended visual checks for OutFLANK, we observe that the relationship between the distribution predicted by OutFLANK and the empirical distribution of F_{ST} is not as good for the three-spined stickleback (left) as for the nine-spined stickleback (right). In particular, the density under the predicted right tail is too strong, indicating that the method is indeed too conservative in this species.

Avoiding methods that induce a species-specific bias has been our strongest concern for this manuscript, so while the reasons for this bias deserves further investigation, we decided to not include the results. In contrast, the two methods that we already have used, i.e. Lositan (assuming the island model) and Bayescan (relaxing the conditions of the island model by allowing for different population sizes and migration rates) are less conservative than OutFLANK, and rely on assumptions that are less sophisticated but that may also reduce any species-specific bias. Indeed, in a comparative framework it might be more useful to have standard models that are “equally bad” for both species than an advanced model that works fine for one species but not for another. While Bayescan was more conservative than Lositan, both methods detected a larger proportion of outlier loci in the three-spined stickleback than in the nine-spined stickleback (Supplementary Table 4). This result is also consistent with the association analysis between environmental variables and SNPs (controlling for population structure), suggesting a stronger environmental contribution to population divergence in the three-spined stickleback. In summary, we believe we have reached a sound conclusion that is not based on F_{st} outlier detection methods alone.

Some detailed remarks:

L60. “phenotypic and genomic adaptive divergence”. See my comments above.

L96-97. Ditto.

L120-121. Ditto.

L125. Interestingly, here plasticity is mentioned, but even if so, the reference is to “adaptive plasticity”. Why would it need to be adaptive? I do not think you have way of inferring this from your data.

L144. “phenotypic and genomic adaptive divergence”. See my comments above – this simply unjustified: you are stretching your interpretations beyond the data.

L219. “adaptive divergence is...” see above.

A7 – We have edited each of these sections to account for the non-adaptive processes that might be at work (see marked changes throughout the manuscript).

L254-256. What about: Rastas P, et al 2016. Genome Biology & Evolution 8:78-93?

A8 – We have now cited this publication since it adds new insights in the genomic synteny between both stickleback species (see **Discussion > paragraph 2 > citation 41**). Still, which genomic features influence population divergence in the nine-spined stickleback remains largely unknown. As a result, the genomic basis of differences in evolutionary versatility between both stickleback species remains to be identified.

L286. “Adapt”. Would respond be more prudent?

A9 – Changed to “diverged” (**Last paragraph of the discussion > first sentence**).

L346. From supplementary methods I found that you used also Arlequin. This came bit as surprise as it was not mentioned here. Why?

A10 – The Arlequin results (which are based on a hierarchical island model) are not shown because they did not consistently reveal a more conservative estimate for the proportion of outlier loci than Lositan (which is based on the classical island model). This was already explained in the Supplementary Methods (**See Supplementary Methods > Genomic signatures of selection > Second paragraph**). Similar to OutFLANK (see answer A6), Arlequin is expected to reduce false positives, in this case by taking into account a hierarchical population structure. The lack of a more conservative estimate than Lositan might indicate that any non-independence of populations for which Arlequin tries to correct is in fact non-hierarchical.

Table 2. I would be more satisfactory to see formal analyses whether PST estimates are in fact significantly higher than FST. It is not straightforward to deduce if this the case from the numbers given in this table.

A11 – We agree that what remains to be shown is which single-trait PST values significantly exceed neutral FST. We therefore quantified Bayesian credible intervals to assess the significance of the difference between single-trait PST and neutral FST (**See methods > Morphological characterisation and analyses and Supplementary Methods > Data Analysis > Phenotypic Differentiation**). The results are added to **Supplementary Table 2**, and are summarized in **Table 2** as the proportion of significant single-trait PST values. This proportion is higher in the three-spined (8 out of 14 traits; 57 %) than in the nine-spined (1 out of 14 traits; 7 %) stickleback. This is compatible with our expectation that phenotypic divergence in the three-spined stickleback cannot be explained by neutral processes alone.

ADDITIONAL CHANGES

A12 – Note that there was a data point missing in **Supplementary Figure 1A**. The reason is that one population of the three-spined stickleback had a heterozygosity value of 0.21, which was out of the range of the Y-axis. The issue is now solved.

A13 – **Supplementary Table 4** was moved from the Supplementary Results (which now have been deleted).

Reviewers' comments:

Reviewer #1 (Remarks to the Author):

I find that the adjustments that are made in this revised ms adequately answers my different comments and questions to the original manuscript. I find this a very nicely executed and written piece of work.

Reviewer #2 (Remarks to the Author):

I identify myself as the reviewer # 2 in the last round. After reading the authors responses and the new version of the manuscript, I appreciate the serious effort they have made to accommodate both reviewers' requests. While I am satisfied with many of the improvements and actions the authors have undertaken, I find myself still worried about the inference about selection and adaptive nature of observed differentiation.

My main concern relates to the inference about outlier loci. As I requested, the authors performed the OutFLANK analyses, which revealed no outliers. The authors argue that this might be because OutFLANK may be too conservative, and the data from the one of the species do not appear to follow (strictly) the chi-distribution expected under null-distribution, rendering the results difficult to interpret. While this may be so, these facts raise – in my mind – serious concerns as OutFLANK has become the 'gold standard' for conducting outlier analyses because the other methods are known to be prone to false positives. In fact, it is a fairly common observation that Bayescan and Lositan in particular yield much more outliers than OutFLANK, many of which may be false positives. For instance, in the supplementary materials (Table 4) it seems that Lositan identified between 7 % (ninespine) and almost 10% (threespine) of the loci as outliers. Bayescan identified far, far less (about 1 %). The idea that 10 % of the genome is under selection seems rather suspicious, but it's a fairly common Lositan result. From the paper it appears also that the conditions for running Lositan were not particularly stringent (for example, q values as per <https://www.ncbi.nlm.nih.gov/pubmed/24655127> were not calculated and fairly liberal p values were used: people often use 0.001 rather than 0.05). Since both Lositan and Bayescan are known to be highly susceptible to false positives when there is spatial autocorrelation or complex demographic histories (as is the case in this study in particular in the case 3-spine stickleback for which more outliers were detected), these seem to me as relevant and serious concerns. This in particular in view that guarding against false positives is something which has become a major issue in genome scan studies (see Bierne et al 2013 <http://onlinelibrary.wiley.com/doi/10.1111/mec.12241/full>, Narum's paper <http://onlinelibrary.wiley.com/doi/10.1111/j.1755-0998.2011.02987.x/full>, and Lotterhos and Whitlock <https://www.ncbi.nlm.nih.gov/pubmed/24655127>).

I was also left to think that even if not using OutFLANK, the authors might have made their approach more stringent by choosing a consensus of loci jointly identified by all three tests (which would have been the most logical thing to do). Instead, the authors decided to define their set of outliers according to the test (Lositan) which identified the largest

number, and is likely to be the most prone to false positives. The authors defend their choice of using all loci identified by Lositan by saying "Since a larger pool of outliers facilitates the comparison of putatively adaptive genomic characteristics, we only proceeded with the LOSITAN SNP matrices to further analyses and compare patterns of neutral and non-neutral population divergence (see below)" (Lines 252-254). Of course, I understand that having more loci would be useful, but if there is a risk many of those are not really under selection then what would be the point of using them?

In short, the main message I am trying to get across is that it may not be crucial exactly which strategy authors follow to control for false positives, but they do need to at least have a strategy.

Minors

I could not find info on which Prior Odds they used in Bayescan, which I think should be added. Foll in the manual suggests that "A value of 10 seems reasonable for the identification of candidate loci within a few hundreds of markers, whereas values up to 10 000 are generally used in the context of genome wide association studies with millions of SNPs when people want to identify only the top candidates.". Anything less than 100 could be considered as a red flag. In a recent paper (Lottheros and Whitlock <https://www.ncbi.nlm.nih.gov/pubmed/24655127>) the authors suggests that using much higher prior odds greatly reduces false positive rates.

L24. This is strong statement - strictly speaking there is no phenotypic evidence for adaptation.

L28. What exactly are the different strategies?

L63 Both -> The two

L91-92. This merely a reflection, but still perhaps relevant one. The way how the fish are caught to obtain the density estimates may influence the relative abundance estimates. If the other species is easier to catch (e.g. goes more readily in trap, shows higher propensity to shoaling, or exhibits habitat preference and behavior making more vulnerable for catching), this could lead apparent difference in abundance?

L153. See above. I think claiming adaptive phenotypic divergence is bit strong here. Also, given the concern about false positives, the inference about adaptive genetic differentiation can be called into question.

L154-155. The stronger propensity for adaptation in threespine is an interesting idea, but until the false positive issue is sorted out, this can be called into question. It is also strange that this would be the case as the genetic drift is claimed to be stronger in the threespine (L156), which contradicts the idea higher propensity for adaptive differentiation – genetic drift is expected reduce the efficiency of selection and work against adaptation. Also this raises the concern that the outliers could be false positives owing to demographic effects.

Reviewers' comments:

Reviewer #1 (Remarks to the Author):

I find that the adjustments that are made in this revised ms adequately answers my different comments and questions to the original manuscript. I find this a very nicely executed and written piece of work.

A1 – Thanks!

Reviewer #2 (Remarks to the Author):

I identify myself as the reviewer # 2 in the last round. After reading the authors responses and the new version of the manuscript, I appreciate the serious effort they have made to accommodate both reviewers' requests. While I am satisfied with many of the improvements and actions the authors have undertaken, I find myself still worried about the inference about selection and adaptive nature of observed differentiation.

My main concern relates to the inference about outlier loci. As I requested, the authors performed the OutFLANK analyses, which revealed no outliers. The authors argue that this might be because OutFLANK may be too conservative, and the data from the one of the species do not appear to follow (strictly) the chi-distribution expected under null-distribution, rendering the results difficult to interpret. While this may be so, these facts raise – in my mind – serious concerns as the OutFLANK has become the 'gold standard' for conducting outlier analyses because the other methods are known to be prone to false positives. In fact, it is a fairly common observation that Bayescan and Lositan in particular yield much more outliers than OutFLANK, many of which may be false positives. For instance, in the supplementary materials (Table 4) it seems that Lositan identified between 7 % (ninespine) and almost 10% (threespine) of the loci as outliers. Bayescan identified far, far less (about

1 %). The idea that 10 % of the genome is under selection seems rather suspicious, but it's a fairly common Lositan result. From the paper it appears also that the conditions for running Lositan were not particularly stringent (for example, q values as per <https://www.ncbi.nlm.nih.gov/pubmed/24655127> were not calculated and fairly liberal p values were used: people often use 0.001 rather than 0.05). Since both Lositan and Bayescan are known to be highly susceptible to false positives when there is spatial autocorrelation or complex demographic histories (as is the case in this study in particular in the case 3-spine stickleback for which more outliers were detected), these seem to me as relevant and serious concerns. This is in particular in view that guarding against false positives is something which has become a major issue in genome scan studies (see Bierne et al 2013 <http://onlinelibrary.wiley.com/doi/10.1111/mec.12241/full>, Narum's paper <http://onlinelibrary.wiley.com/doi/10.1111/j.1755-0998.2011.02987.x/full>, and Lotterhos and Whitlock <https://www.ncbi.nlm.nih.gov/pubmed/24655127>).

I was also left to think that even if not using OutFLANK, the authors might have made their approach more stringent by choosing a consensus of loci jointly identified by all three tests (which would have been the most logical thing to do). Instead, the authors decided to define their set of outliers according to the test (Lositan) which identified the largest number, and is likely to be the most prone to false positives. The authors defend their choice of using all loci identified by Lositan by saying "Since a larger pool of outliers facilitates the comparison of putatively adaptive genomic characteristics, we only proceeded with the LOSITAN SNP matrices to further analyses and compare patterns of neutral and non-neutral population divergence (see below)" (Lines 252-254). Of course, I understand that having more loci would be useful, but if there is a risk many of those are not really under selection then what would be the point of using them?

In short, the main message I am trying to get across is that it may not be crucial exactly which strategy authors follow to control for false positives, but they do need to at least have a strategy.

A2 – We accept the criticism of the reviewer and agree that stronger control for false positives was needed. Following Lotterhos & Whitlock (2015), we have therefore applied FDR control at a q-value of 0.05 for all outlier detection methods (Bayescan, Lositan and Arlequin). In addition, we also still apply this procedure 3 times, and only considered loci that are thrice significant at this level as real outliers. As a result, the rate of outliers dropped dramatically to 0.5-2 % for the three-spined stickleback, and 0.08-0.22 % for the nine-spined stickleback, depending on the method. Yet, the support for the main conclusion, the stronger propensity for adaptation in the three-spined stickleback, has become considerably stronger.

We do not follow the suggestion of the reviewer to take the consensus of all three methods. First, as before, we want to show that the main conclusion is supported by each method independently (Supplementary Table 4). As the reviewer points out, the migration-drift-selection balance might differ between species, and therefore reporting each method separately is important to maximize transparency with respect to the sensitivity of the

underlying models. In addition, by taking the consensus of all three methods, the actual FDR becomes unknown, which we believe is not desirable, and may enhance the risk of missing true positives.

As we have shown convincingly in the very first submission (Supplementary Results), the patterns of neutral and non-neutral divergence are quantitatively and qualitatively very similar across methods. For all downstream analyses, we therefore still choose to only present a single method. We here opted for Bayescan, because this method resulted in the smallest difference in the proportion of outliers between both species. This is the most conservative choice given the scope of the study.

Because of the stricter FDR control and associated shift in neutral F_{st} , various results changed slightly, and therefore almost all figures and tables have been updated. In addition, a number of analyses became redundant because of the very low number of outlier loci in the nine-spined stickleback. Yet, since our overall conclusions remain, the main text has not changed a lot (see track changes). The largest change is that we no longer find common outliers between the two species, and therefore we have deleted the former Supplementary Table 6 and corresponding conclusions.

We thank the reviewer for the advice, which has enhanced the quality of our analyses and makes the conclusions more sound. We have now confirmed our conclusion of a stronger propensity for adaptation in the three-spined stickleback across a wide range methods and cut-off values. This conclusion is also compatible with the stronger phenotypic divergence and the stronger correlation between outlier loci and environmental variables in this species. The complete lack of outlier loci obtained with OutFlank remains curious, but is in any case highly incompatible with any other test.

Minors

I could not find info on which Prior Odds they used in Bayescan, which I think should be added. Foll in the manual suggests that “A value of 10 seems reasonable for the identification of candidate loci within a few hundreds of markers, whereas values up to 10 000 are generally used in the context of genome wide association studies with millions of SNPs when people want to identify only the top candidates.”. Anything less than 100 could be considered as a red flag. In a recent paper (Lottheros and Whitlock <https://www.ncbi.nlm.nih.gov/pubmed/24655127>) the authors suggests that using much higher prior odds greatly reduces false positive rates.

A3 – We have now specified that the prior odds was set to 100 (Supplement L239). Since our previous analyses were based on a prior odds of 10, we performed all Bayescan runs again and updated the results.

L24. This is strong statement - strictly speaking there is no phenotypic evidence for adaptation.

A4 – We have modified the sentence to avoid this claim.

L28. What exactly are the different strategies?

A5 – We have elaborated on that in the Discussion.

L63 Both -> The two

A6 – Changed accordingly.

L91-92. This merely a reflection, but still perhaps relevant one. The way how the fish are caught to obtain the density estimates may influence the relative abundance estimates. If the other species is easier to catch (e.g. goes more readily in trap, shows higher propensity to shoaling, or exhibits habitat preference and behavior making more vulnerable for catching), this could lead apparent difference in abundance?

A7 – This is a theoretical possibility, but the physical structure of the sites is essentially so similar that catch bias can certainly not explain the discrepancy between freshwater sites and brackish water sites alone. In the supplementary methods (L132) we have already mentioned previously that “Given that all sites are shallow (i.e., < 75 cm depth: Supplementary Table 1), vegetated and narrow (i.e., < 3 meter), any catch bias across sites and species was probably negligible.” Along each river stretch, there is also variation in vegetation and water depth, which likely averages out the microhabitat differences. Temporal replication further adds confidence to the observed differences in relative abundance.

L153. See above. I think claiming adaptive phenotypic divergence is bit strong here. Also, given the concern about false positives, the inference about adaptive genetic differentiation can be called into question.

A8 – As for answer A4, we have edited the sentence in order to avoid the claim of adaptive phenotypic divergence. Yet, it is now very clear that reducing false positives for all three outlier detection methods leads to a higher rate of outlier loci in the three-spined stickleback than in the nine-spined stickleback (Supplementary Table 4).

L154-155. The stronger propensity for adaptation in threespine is an interesting idea, but until the false positive issue is sorted out, this can be called into question. It is also strange that this would be the case as the genetic drift is claimed to be stronger in the threespine (L156), which contradicts the idea higher propensity for adaptive differentiation – genetic drift is expected reduce the efficiency of selection and work against adaptation. Also this raises the concern that the outliers could be false positives owing to demographic effects.

A9 – See A2 and A8 – the false positive issue is now sorted out and only made this conclusion stronger. It indeed contradicts existing theory on the role of genetic drift. While this warrants further investigation, we would like to emphasize that none of the three-spined stickleback populations is very small (see Supplementary Table 1). While genetic drift should act stronger in the three-spined stickleback than in the nine-spined stickleback, it is perhaps not at a level that it can strongly hinder selection. Furthermore, our analyses suggest that both processes may truly coincide in the three-spined stickleback. Therefore it might be harder in this species to isolate what is pure drift and what is pure selection.

REVIEWERS' COMMENTS:

Reviewer #2 (Remarks to the Author):

I am glad to see that authors have taken home the earlier criticism and re-evaluated the conclusions based on new analyses. As a result, the number of outliers detected dropped dramatically indicating that the earlier results were strongly influenced by false positives. While I think the paper is now much more sound than earlier, I am still slightly concerned about few issues.

First, the number of outliers (i.e. loci indicated to be under selection) is now very low, and the difference in the proportion of outliers between species is very small (0.5 – 2% vs 0.08 – 0.22%) making one to wonder about the biological significance of this difference. This especially in the view of the difficulty of sorting out false positives.

Second, I am not totally convinced by authors argument not use the consensus of three different methods when calling the outliers. Namely, while it may be true that FDR becomes unknown for such an overall result, using the consensus should be most conservative way of calling the outliers. It would have been nice to know if the conclusions would have been changed if such a call had been made. Now one is left to wonder if the authors' reluctance of using consensus call hides the fact that the results are sensitive to outlier method used. Clearly, were the results supporting the inference the same as now as they would be if only outliers detected by the three different methods, this would be (to my mind) very convincing evidence for robustness of the results.

Third, authors' argument about reliability of abundance estimates is understandable, but in absence of additional evidence, it is very much inferential. I do not think that the catch bias can be excluded. For instance, temporal replication may be just replication of the existing (unidentified) biases.

To sum up, I think this is a well-written and interesting study, but given the low number of outliers and especially the small differences between species, together with some remaining methodological concerns, I am left to wonder if we actually learn very much from this study?

REVIEWER #2

I am glad to see that authors have taken home the earlier criticism and re-evaluated the conclusions based on new analyses. As a result, the number of outliers detected dropped dramatically indicating that the earlier results were strongly influenced by false positives. While I think the paper is now much more sound than earlier, I am still slightly concerned about few issues.

First, the number of outliers (i.e. loci indicated to be under selection) is now very low, and the difference in the proportion of outliers between species is very small (0.5 – 2% vs 0.08 - 0.22%) making one to wonder about the biological significance of this difference. This especially in the view of the difficulty of sorting out false positives.

A1: The percentage difference in outliers between the species is indeed small, but is consistent across outlier detection methods, while the order of magnitude is large (2.5 to 10 times larger; i.e. 283 vs 21 outlier snps for Lositan; 66 vs 8 outlier snps for Arlequin; 70 vs 22 outlier snps for Bayescan). Yet, there are no rules whether as to express the discrepancy between the species on a percentage scale or on an absolute scale, and we therefore present both ways. Key results of Supplementary Table 4 have therefore been added to the main text. We have also edited the discussion (**paragraph on genomics**) to emphasize that further study will be required to evaluate the true effect size of the observed discrepancy between the species, e.g. by analysing the structure and function of target genomic regions.

After the previous review round, we no longer share the continued concern of reviewer 2 about false positives. We have applied an FDR which is widely accepted, and in our opinion it is unnecessary to maintain a stricter FDR given the exploratory and comparative character of the study, where it is also important to prevent false negatives (see Narum & Hess 2011). While it is logical that stricter FDR control ultimately reduces the percentage difference between the species, we observe that the discrepancy in orders of magnitude increases with stricter FDR. The discrepancy also remains when combining methods (see **answer A2**).

Second, I am not totally convinced by authors argument not use the consensus of three different methods when calling the outliers. Namely, while it may be true that FDR becomes unknown for such an overall result, using the consensus should be most conservative way of calling the outliers. It would have been nice to know if the conclusions would have been changed if such a call had been made. Now one is left to wonder if the authors' reluctance of using consensus call hides the fact that the results are sensitive to outlier method used. Clearly, were the results supporting the inference the same as now as they would be if only outliers detected by the three different methods, this would be (to my mind) very convincing evidence for robustness of the results.

A2: Here is the consensus analysis between Bayescan-Arlequin, Bayescan-Lositan and Arlequin-Lositan:

14 of 70 bayescan outliers are also arlequin outliers for the three-spined stickleback

1 of 22 bayescan outliers is also an arlequin outlier for the nine-spined stickleback

43 of 70 bayescan outliers are also lositan outliers for the three-spined stickleback

4 of 22 bayescan outliers are also lositan outliers for the nine-spined stickleback

all 66 arlequin outliers are also lositan outliers for the three-spined stickleback

all 8 arlequin outliers are also lositan outliers for the nine-spined stickleback

The intersection between each two methods always contain 8-14 times more outliers for the three-spined stickleback than for the nine-spined stickleback, exceeding the discrepancy of the most conservative independent method (= Bayescan). Combining methods hence does not lead to a more conservative estimate of the discrepancy between the species. We therefore believe it is not particularly useful to present these consensus results. Indeed, the analyses can be done in many different ways, and we really prefer to keep our methods simple and transparent. It is also important to realise that outlier detection methods are not the holy grail of calling genomic signatures of selection. In particular, once the nine-spined stickleback genome also becomes available, there will be opportunities for more sophisticated strategies to identify adaptive genomic regions, such as sliding window analyses and hidden markov models, and to compare those regions among the two species.

Third, authors' argument about reliability of abundance estimates is understandable, but in absence of additional evidence, it is very much inferential. I do not think that the catch bias can be excluded. For instance, temporal replication may be just replication of the existing (unidentified) biases.

A3: We agree that this is inferential, but we are confident that IF three-spined sticklebacks are easier to catch than nine-spined sticklebacks at the freshwater sites (which we strongly doubt), the bias would be too small to explain the systematic difference with the brackish water sites. For transparency, we have added an additional

statement in Figure legend 1: “Sampling bias unlikely explains this result, because standard catches were done with a hand net at sites with little escape opportunity (see Supplementary Methods).” This will help the reader to appreciate the potential methodological limitations.

To sum up, I think this is a well-written and interesting study, but given the low number of outliers and especially the small differences between species, together with some remaining methodological concerns, I am left to wonder if we actually learn very much from this study?

A4: It should not be overlooked that we have opened the results with a rather spectacular effect size at the phenotypic level (strong P_{st} and ratio of $P_{st}/neutral F_{st}$ in the three-spined stickleback vs weak values in the nine-spined stickleback), including for traits with a known genetic basis. Some of these traits (dorsal spine length, body depth, gill raker length) also show a remarkable parallelism between the two species. Thus, we can certainly learn a lot from this comparison. Our results of more outliers in the three-spined stickleback provide a valid explanation for the phenotypic patterns. It is true that the effect size of this result remains unknown (see **answer A1**), but it would be very surprising if there would be no such effect. Alternative explanations (plastic and transgenerational effects) remain possible, and have been fully acknowledged in the manuscript.